# Awareness of the role of general practitioners in primary care among outpatient populations: evidence from a cross-sectional survey of tertiary hospitals in China

Xiaojun Liu,[1,2] Anran Tan,[1] Samuel D Towne Jr,[3] Zhaoxun Hou,[1] Zongfu Mao[1,2]

XL and AT contributed equally.

[1]School of Health Science, Wuhan University, Wuhan, China
[2]Global Health Institute, Wuhan University, Wuhan, China
[3]School of Health Science, Texas A&M University, College Station, Texas, USA

**Correspondence to**
Professor Zongfu Mao;
zfmao@whu.edu.cn

## ABSTRACT

**Objective** General practitioners (GPs) are highly underutilised in China with many patients going directly to hospitals when seeking routine care. Multiple countries around the world have successfully used GPs in routine care, and as such, China may benefit from the use of GPs. This study examines the status of, and factors associated with, knowledge related to GPs among outpatient populations from China's tertiary hospitals.

**Design** This is a cross-sectional survey study.

**Study setting and participants** The questionnaires were completed by 565 outpatients from four tertiary hospitals in China during 2016. Convenience sampling on different floors and throughout the outpatient building was carried out.

**Primary outcome measures** We used the logistic regression models to identify GP-related knowledge among different populations.

**Results** Overall, 50.27% of respondents said they had never heard of GPs. This was also true among females (adjusted OR (AOR)=1.57, 95% CI 1.43 to 2.71), older adults (AOR$_{46-65}$=1.61, 95% CI 1.39 to 2.98; AOR$_{>65}$=2.01, 95% CI 1.62 to 3.59), those with lower education level (AOR$_{Bachelor's\ degree}$=0.61, 95% CI 0.20 to 0.81; AOR$_{\geq Master's\ degree}$=0.49, 95% CI 0.23 to 0.76), rural residents (AOR=1.51, 95% CI 1.35 to 2.82) and those with chronic disease (AOR$_{without\ chronic\ disease}$=0.61, 95% CI 0.22 to 0.71). What is more, less than one-in-ten (9.03%) outpatients were able to accurately describe what a GP was, with less than 30% accurately describing a GP among those receiving GPs' services.

**Conclusions** Outpatients who could have received less costly health services from GPs in primary medical institutions were more likely to choose costlier specialist physicians in tertiary hospitals, which is likely linked to limited knowledge about GPs. Policy makers should invest in outreach efforts to improve public awareness of GPs, while at the same time conducting continued surveillance of these efforts to evaluate progress towards this goal.

## Strengths and limitations of this study

► This is the first study that represents the current status of, and factors associated with, outpatients' knowledge related to general practitioners (GPs) in China.

► The survey was conducted among outpatient populations from Chinese tertiary hospitals who could have received less costly health services from GPs in primary medical institutions.

► Potential bias was reduced given the questionnaires were completed anonymously.

► The geographic scope of the study was limited and as such may not be representative of other areas throughout China.

► Non-response bias was not assessed, as only those agreeing to participate were included in study analyses.

and waste in healthcare resources,[1–3] is a critical issue that must be addressed. Medical workers in tertiary comprehensive hospitals are overwhelmed by patients, leading to long wait times for patients, and therefore increasing the potential for negative experiences among patients.[4 5] At the same time, primary care medical facilities may not be equipped with advanced medical resources, and as such, this may drive health-seeking behaviour towards tertiary comprehensive hospitals rather than utilisation of potentially more appropriate primary care settings.[6 7] Thus, misuse of medical resources continues to be a major issue throughout China.[1 4 6 8–10]

In order to achieve the goal of delivering high-quality medical resources in primary care settings in an effective way, the Chinese government is currently promoting the establishment and implementation of a hierarchical treatment system, aiming to accomplish a more efficient allocation of treatment

## INTRODUCTION

Inappropriate allocation of current health resources in China, in particular the shortage

by promoting primary care with a two-way referral mechanism, within acute and chronic patients while integrating centralised efforts and at the same time strengthening more local efforts as well. The first step in establishing a hierarchical treatment system is to develop a national general practitioner (GP) system. The GPs are a critical factor in the national GP system, whose main role is to provide patients with health services in the 'integration of six aspects' in primary care facilities.[11–13] 'Integration of six aspects' in primary care facilities refers to the community health service network system which integrates community prevention, healthcare, medical treatment, rehabilitation, health education and family planning technology guidance. The comprehensive function of the concept meets various requirements in healthcare.[12 13] However, the establishment and development of the national GP system not only requires the government's relative policy guidance, but also demands the recognition from both doctors and patients to effectively implement and carry out this change.

In China, GPs are typically referred to as family practitioners (FPs). However, GPs (or FPs) in China are not the same as primary care physicians (PCPs). In China, PCPs are also referred to as 'barefoot doctors' (in Chinese 'Chijiao doctors' or 'Tongke practitioners'). PCPs (or barefoot doctors) are given basic training in Western disease control and traditional Chinese medicine or ethnic medicine and typically have limited technical skills or medical capability.[14] China developed the national GP policy to provide general practice based primary care, and this has been implemented to varying extents across the country. The Chinese government continues to train qualified GPs with the goal of replacing existing PCPs (or barefoot doctors).

Patients may benefit from receiving care from GPs in primary care facilities with potential referral to specialist care in hospital settings as necessary. However, many seek specialist physicians in tertiary comprehensive hospitals directly, which is potentially more expensive and possibly more inconvenient given longer distances travelled by some. This phenomenon may be caused by a lack of knowledge of GPs. Inadequate research exists on the role that inadequate public awareness and knowledge of GPs may play in health-seeking behaviour. Thus, we aimed to focus on outpatients seeking health services in top tier provincial tertiary hospitals in China. This line of inquiry can contribute to policy recommendations and accelerating the promotion of the national GP system and the hierarchical treatment system with the goal of a more efficient allocation health resources in China.

## MATERIALS AND METHODS
### Participants
Outpatients with a registered physician in provincial tertiary hospitals were invited to take part in a survey, while those referred from other health service institutions were excluded.

### Study procedure and data collection
We conducted a cross-sectional study using surveys in four of the top rated (eg, largest hospitals with high levels of resources and high-quality care) provincial tertiary hospitals in Jiangxi Province. Survey respondents included 150 people at each hospital, with a total of 600 respondents in 2016. Convenience sampling and spatial sampling (ie, targeting different floors and directions to improve recruitment of potential participants) were used to collect the data. Questionnaires were given to the registered outpatients who were waiting to see a doctor in different floors and different directions of the outpatient building. Those who qualified (ie, including those with a registered physician in a provincial tertiary hospital, while excluding those referred from another health service institutions) were informed of the study and asked if they were willing to participate establishing informed consent. The targeted individuals were asked to answer the questionnaires independently and anonymously. Participants were interviewed by trained interviewers if they requested assistance completing the survey (eg, outpatients who were illiterate). Lastly, completed questionnaires were checked by qualified investigators (ie, graduate students specifically trained to carry out data collection for this study) to ensure the completeness of the questionnaires with immediate follow-up with participants needing further information, as needed.

### Survey tool
The questionnaire was designed based on the *Introduction to General Practice*,[14] and with the help of scholars familiar with the topic with final review by clinical experts in the field (ie, university hospital instructors of GPs). Surveys were pretested prior to study implementation to ensure that potential participants understood the wording and meaning of study questionnaires. The survey instrument was focused on participants' knowledge of GPs. Specifically, the questionnaire inquired about: (1) social demographic characteristics (table 1) including participants' sex, age, education level, average monthly income, place of residence, type of healthcare insurance and chronic disease status (self-reported yes/no); (2) participants' knowledge of GPs: consisting of six questions (table 2).

The Cronbach's alpha coefficient of this questionnaire was 0.91, indicating that the survey tool for the present study had good internal reliability. On average, participants completed the survey in less than 3 min.

### Quality control
Targeted individuals who completed survey questions independently were asked to answer the questionnaires anonymously based on their own knowledge and understanding of GPs, with no mention of personal details to avoid bias due to induced prompts by the investigators. For those who were unable to complete survey questions independently, we employed a rigorously trained and qualified staff of investigators to interview them face to face using a consistent and clear explanation of the

| Table 1 | Demographic information of the survey participants (n=565) | |
|---|---|---|
| | **Frequency** | **Percentage** |
| **Sex** | | |
| Male | 307 | 54.34 |
| Female | 258 | 45.66 |
| **Age** | | |
| ≤25 | 83 | 14.69 |
| 26–45 | 187 | 33.10 |
| 46–65 | 196 | 34.69 |
| >65 | 99 | 17.52 |
| **Education level** | | |
| ≤Junior high school | 397 | 70.27 |
| Bachelor's degree | 102 | 18.05 |
| ≥Master's degree | 66 | 11.68 |
| **AMI (RMB)** | | |
| <3000 | 181 | 32.04 |
| 3000–4999 | 199 | 35.22 |
| 5000–6999 | 106 | 18.76 |
| ≥7000 | 79 | 13.98 |
| **Place of residence** | | |
| City | 329 | 58.23 |
| Countryside (rural) | 236 | 41.77 |
| **Types of healthcare insurance** | | |
| No insurance | 67 | 11.86 |
| Yes, NRCMS | 303 | 53.63 |
| Yes, MISUR | 195 | 34.51 |
| **Physical condition** | | |
| With chronic disease | 176 | 31.15 |
| Without chronic disease | 389 | 68.85 |

AMI, average monthly income; MISUR, medical insurance system for urban residents; NRCMS, new rural cooperative medical system.

purpose and significance of the study before the survey was conducted while at the same time answering questions raised by potential participants as they were encountered. The returned questionnaires were checked in a timely manner with incomplete surveys given back to outpatients to complete. In cases of refusal, surveys were excluded so only complete surveys were included. Double entry and logical verification of the questionnaires was done to ensure the accuracy of the data.

## Statistical analysis

Statistical Package for the Social Sciences (SPSS) V.22.0 for Windows (SPSS Inc.) was employed to run all statistical analysis. A p value <0.05 was considered as statistically significant. Initial descriptive analysis summarised participants' social demographic characteristics and knowledge of GPs, with frequencies and proportions presented in

| Table 2 | Respondents' knowledge of the GPs | |
|---|---|---|
| **Items** | **Frequency** | **Percentage** |
| **Have you ever heard of a GP?** | | |
| Yes | 281 | 49.73 |
| No | 284 | 50.27 |
| **Have you received health services from GP(s)?** | | |
| Yes | 153 | 27.08 |
| No | 66 | 11.68 |
| Do not know | 346 | 61.24 |
| **How would you rate a GP's technical capability?** | | |
| Good | 64 | 11.33 |
| Poor | 149 | 26.37 |
| Do not know | 352 | 62.30 |
| **Do you think a GP is the same as a PCP?** | | |
| Yes | 516 | 91.33 |
| No (correct answer) | 49 | 8.67 |
| **Do you think a GP is the same as a FP?** | | |
| Yes (correct answer) | 88 | 15.58 |
| No | 477 | 84.42 |
| **Do you think that GP should handle CFOD?** | | |
| Yes (correct answer) | 379 | 67.08 |
| No | 186 | 32.92 |
| Total | 565 | 100.00 |

CFOD, common and frequently occurring diseases; FP, family practitioner; GP, general practitioner; PCP, primary care physician.

tables 1 and 2, figure 1. Univariate and multivariate analysis were performed by using binary logistic regression analysis to identify the main factors associated with outpatients' knowledge related to GPs. Both the crude OR and adjusted OR (AOR) with associated 95% CIs are reported in table 3.

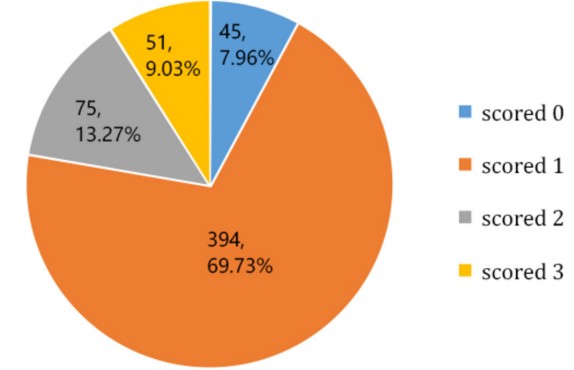

**Figure 1** Respondents' general awareness rate of knowledge towards the general practitioners.

**Table 3** Knowledge about GPs among different groups

| Demographics | No<br>n=284 (50.27%) | Yes<br>n=281 (49.73%) | Crude<br>OR (95% CI) | Adjusted<br>OR (95% CI) |
|---|---|---|---|---|
| **Sex** | | | | |
| Male | 136 (44.30) | 171 (55.70) | – | – |
| Female | 148 (57.36) | 110 (42.64) | 1.69 (1.21 to 2.36)*** | 1.57 (1.43 to 2.71)*** |
| **Age** | | | | |
| ≤25 | 35 (42.17) | 48 (57.83) | – | – |
| 26–45 | 86 (45.99) | 101 (54.01) | 1.17 (0.47 to 2.03) | 1.02 (0.53 to 2.16) |
| 46–65 | 105 (53.57) | 91 (46.43) | 1.58 (0.97 to 3.01) | 1.61 (1.39 to 2.98)* |
| >65 | 58 (58.59) | 41 (41.41) | 1.94 (1.73 to 4.15)* | 2.01 (1.62 to 3.59)* |
| **Education level** | | | | |
| ≤Junior high school | 218 (54.91) | 179 (45.09) | – | – |
| Bachelor's degree | 41 (40.20) | 61 (59.80) | 0.55 (0.36 to 0.85)** | 0.61 (0.20 to 0.81)** |
| ≥Master's degree | 25 (37.88) | 41 (62.12) | 0.50 (0.31 to 0.87)** | 0.49 (0.23 to 0.76)** |
| **AMI (RMB)** | | | | |
| <3000 | 109 (60.22) | 72 (39.78) | – | – |
| 3000–4999 | 93 (46.73) | 106 (53.27) | 0.58 (0.56 to 0.95)** | 0.52 (0.49 to 1.53) |
| 5000–6999 | 51 (48.11) | 55 (51.89) | 0.61 (0.50 to 0.91)** | 0.67 (0.44 to 1.67) |
| ≥7000 | 31 (39.24) | 48 (60.76) | 0.43 (0.34 to 0.93)** | 0.54 (0.29 to 1.01) |
| **Place of residence** | | | | |
| City | 148 (44.98) | 181 (55.02) | – | – |
| Countryside (rural) | 136 (57.63) | 100 (42.37) | 1.66 (1.19 to 2.33)** | 1.51 (1.35 to 2.82)** |
| **Types of healthcare insurance** | | | | |
| No insurance | 35 (52.24) | 32 (47.76) | – | – |
| Yes, NRCMS | 155 (51.16) | 148 (48.84) | 0.96 (0.83 to 1.31) | 0.92 (0.76 to 1.73) |
| Yes, MISUR | 94 (48.21) | 101 (51.79) | 0.85 (0.66 to 1.17) | 0.86 (0.53 to 1.03) |
| **Physical condition** | | | | |
| With chronic disease | 108 (61.36) | 68 (38.64) | – | – |
| Without chronic disease | 176 (45.24) | 213 (54.76) | 0.52 (0.36 to 0.75)*** | 0.61 (0.22 to 0.71)*** |

*P<0.05, **P<0.01, ***P<0.001.
AMI, average monthly income; GP, general practitioner; MISUR, medical insurance system for urban residents; NRCMS, new rural cooperative medical system.

## RESULTS
### Descriptions of sample demographic
As table 1 shows, the final sample of participants was 565 with an effective survey response rate of 94.17%, of which most were men (54.34%) and urban residents (58.23%). The largest age group was 46–65 (34.69%) followed by those aged 26–45 (33.10%). Most individuals had average monthly incomes of less than 5000 RMB with 35.22% of those making 3000–4999 RMB and 32.04% of those making <3000 RMB. Nearly 90% reported having medical insurance with 53.63% having the new rural cooperative medical care and 34.51% having medical insurance system for urban residents. Nearly a third (31.15%) had at least one chronic disease.

### The status of outpatients' knowledge related to GPs
Table 2 illustrates the current status of knowledge related to GPs among outpatients from tertiary comprehensive hospitals. Overall, less than half of the subjects (49.73%) had ever heard of a GP.

Most outpatients were unable to give the correct answers on the basic conceptual knowledge towards GPs. However, there were 379 patients (67.08%) who were able to correctly identify that GP's main focus was on common and frequently occurring diseases. We also produced a figure to highlight correct answer choices where each correct answer contributes one point in three questions and the total score is 3. The results in figure 1 show that there were 45 outpatients (7.96%) who scored

0 points, 394 outpatients (69.73%) who scored 1 point, 75 outpatients (13.27%) who scored 2 points, and less than one-in-ten (n=51, 9.03%) outpatients who correctly answered all questions (scored 3).

### Analysis of knowledge towards GPs by population subgroups

To assess potential differences in knowledge levels towards GPs in different populations, we identified whether patients had ever heard of a GP or not. Univariate and multivariate analysis were performed by using binary logistic regression models to identify the knowledge level about GPs in different groups. The results are shown in table 3. In terms of the knowledge of GPs, both univariate and multivariate analysis showed that female outpatients were more deficient in knowledge of GPs (AOR=1.57, 95% CI 1.43 to 2.71) as compared with males. When compared with younger individuals aged 25 or younger, those aged 46–65 (AOR=1.61, 95% CI 1.39 to 2.98) and those aged 65 or above (AOR=2.01, 95% CI 1.62 to 3.59) were more deficient in knowledge of GPs. Those with a Bachelor's degree (inclusive of having a vocational degree) were less likely to be deficient in knowledge of GPs (AOR=0.61, 95% CI 0.20 to 0.81) and those with a Master's degree (AOR=0.49, 95% CI: 0.23 to 0.76) were also less likely to be deficient in knowledge of GPs. Rural residents were also more likely to be deficient in knowledge of GPs (AOR=1.51, 95% CI 1.35 to 2.82) as compared with urban-dwelling adults. Further, those without chronic disease were less likely to be deficient in knowledge of GPs (AOR=0.61, 95% CI 0.22 to 0.71).

### DISCUSSION

GPs may be most appropriate to care for non-emergency routine medical services, yet awareness of the role or GPs is broadly lacking. Due to China's inverted triangle of health resource allocation, the majority of patients in need of health services rarely consider primary care medical facilities as their first choice, which may be a misalignment of what could be properly allocated medical treatment.[1 3 4 15–18] Therefore, many of the outpatients seeking care in tertiary comprehensive hospitals may be able to seek appropriate care from GPs in primary care healthcare institutions. This paper represents the current status of, and factors associated with, outpatients' knowledge related to GPs in China.

According to the *2016 Statistical Yearbook of China's National Health and Family Planning Commission,*[19] the total number of licensed GPs and assistant GPs was just 5.3% of the total number of physicians in China. The General Office of the State Council promulgated the *Opinion on Promoting the Construction of Hierarchical Medical Treatment System,*[20] which aims to achieve a goal of having two to three qualified GPs for every 10 000 citizens. However, the average number of GPs per 10 000 citizens in China was just 1.27 overall, with 1.71 per 10 000 in more developed provinces in the eastern coastal areas, with 0.91 and 0.99 per 10 000 in less developed areas in central and western

regions, respectively. Jiangxi Province is a typical central province in China with 0.54 GPs per 10 000 permanent residents in 2016, among which, most of the licensed GPs were specialists, leaving few who were truly trained by the standard procedures of GPs providing non-specialist care.[21] Moreover, nearly half of the GPs did not work in primary care facilities, giving way to a large gap in primary care facilities which desperately need GPs.[21 22]

We believe that the establishment on the national GPs system should not only rely on administrative government departments, but also seek buy-in from medical staff, the public (the potential patients) and other key stakeholders. Our study indicates that serious deficits exist in terms of knowledge about GPs among outpatients seeking care in tertiary comprehensive hospitals. Therefore, we recommend that policy makers may use this evidence to take action by targeting awareness campaigns that highlight the role of GPs, the national GPs system and the relative advantages of the national GPs system to the public.

Specific content may cover the function of primary healthcare and general medical treatment in order to raise public awareness of GPs. This may take a variety of stakeholders and media including: social media (eg, WeChat and Weibo) highlighting simple videos concerning the role and importance of GPs; medical workers who may explain relevant roles of GPs to patients; medical schools encouraging students to consider becoming GPs and/ or educating them about how to discuss the importance of GPs in primary care. In terms of tailoring education and awareness campaigns to particularly at-risk groups, our findings suggest that females, older adults, those with lower education, rural residents and those that did not have a chronic disease were most likely to have serious deficits in terms of knowledge about GPs.

### Limitations

Given the target subjects in this study were already outpatients in tertiary comprehensive hospitals, the results may not necessarily generalisable to the general population in China. In addition, the geographic scope of the study was limited and as such may not be representative of other areas throughout China. Further, recall bias may have affected survey responses. Moreover, non-response bias was not assessed, as only those agreeing to participate were included in study analyses. Given the pragmatic nature of data collection in hospital settings, we did not track how many surveys were done through face-to-face interviews versus those where individuals completed the surveys by themselves or those where certain questions were missing, and participants were asked to complete missing items.

### CONCLUSIONS

As an essential factor of the national GP system, GPs can serve as the primary provider of basic medical services for many individuals. However, Chinese outpatients from tertiary comprehensive hospitals have limited knowledge

about GPs. This is most evident among those who were female, older adults, those with lower education, rural residents and those without chronic disease. We recommend that policy makers target investment in public education programmes to raise awareness of GPs and their role. Further, continued surveillance must be carried out in order to identify whether success in achieving these targets is met over time.

**Acknowledgements** We thank all the teachers, students and the research participants who took part in the data collection.

**Contributors** XL and ZM conceived the study; XL collected data and conducted the data analysis; XL, ZH and AT drafted the paper; ZM, SDT and XL revised the manuscript. All authors read and approved the final manuscript. XL and ZM are guarantors of the paper.

**Funding** This work was performed as part of the Texas A&M University Health Science Center–Wuhan University Global Health Research Partnership. This study was also supported by an internal grant from Wuhan University (grant number: S2017801593).

**Disclaimer** The funding body was not involved in study design, data collection, data analysis or the writing of the article.

**Competing interests** None declared.

**Patient consent** Obtained.

**Ethics approval** This study was conducted in accordance with the Declaration of Helsinki. Local institutional protocols at Wuhan University were also followed to protect participants' confidentiality, with ethics approval granted by Wuhan University's School of Public Health (IRB number: J-17-2016). Informed consent was obtained from all participants. Hospitals are considered public places in China, and approval from the ethics committees of the hospitals was not required. The Institutional Review Board of Texas A&M University (IRB number: IRB2016-0285D) also reviewed and approved study protocols.

**Provenance and peer review** Not commissioned; externally peer reviewed.

**Data sharing statement** Data are available upon request from the corresponding author. Data requesters are required to provide their research objective, design and ethical approval documents.

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
