## [Reviewer comments · BMJ Open]

ARTICLE DETAILS

TITLE (PROVISIONAL)	Awareness of the role of general practitioners in primary care among outpatient populations: evidence from a cross-sectional survey of tertiary hospitals in China
AUTHORS	Liu, Xiaojun; Tan, Anran; Towne, Samuel; Hou, Zhaoxun; Mao, Zongfu

VERSION 1 – REVIEW

REVIEWER	Martin Roland University of Cambridge UK
REVIEW RETURNED	02-Dec-2017

GENERAL COMMENTS	Some years ago, China developed a national policy to provide general practice based primary care and this has been implemented to varying extents across the country. However, patients still preferentially go to hospital clinics and primary care facilities may be under-utilised. The paper explores some of the reasons why – including that many patients had never heard of a GP and had no idea what they did. To this extent the paper is useful. It's clearly set out and could be published. The survey achieved a very high response rate of 94% (these types of response rate are hardly ever seen outside China). One issue which is not possible to deduce from the paper is whether the patients were clear what they were being asked about. So the reader doesn't know what term was used for GP, Primary Care Physician or Family Practice and whether respondents could have been confused by the terms used. The best way to address this would be to have the questionnaire survey reviewed by someone bilingual in English and Mandarin or whichever local dialect was used. Other than that, the analyses appear sound and the results clearly presented. What for me is missing from the paper is to locate it in the national policy context and, in particular, the local implementation of that policy in Jianxi province – for example, how many GP are there, how evenly are they spread, how long have they been there etc. Other than that, I think this is an interesting and useful paper and could potentially be published.
--

REVIEWER	Jason Ong
-----------------	-----------

	London School of Hygiene and Tropical Medicine Monash University, Australia
REVIEW RETURNED	12-Dec-2017

GENERAL COMMENTS	MAJOR ESSENTIAL CHANGES line 27 - this is your major result that half of respondents have not heard of GP. This may also be influenced by the the Chinese phrase used for GP - did you use a technical term that most lay people would not understand? please clarify how this question was phrased in your methods section. line 92 - what spatial sampling was used? line 94 - it is not clear what type of outpatients has been recruited. this sentence says "general outpatient hall" - can you please expand? line 94 - those who "qualified" - what were your inclusion/exclusion criteria? line 105 - how was chronic disease status defined? line 119 - it is not clear how your multivariate model was built? did you use backward elimination? another technique? what stats package was used? Discussion - more detail is needed regarding how your results compare with other published literature in the Chinese context MINOR ESSENTIAL CHANGES Line 21 - rephrase.. .the study is not about 'GPs in China's tertiary hospitals' - the last part of the sentence refers to outpatient attendees line 23 - add whether the 565 outpatients were sequentially/randomly chosen, from any particular speciality outpatients? line 71 - please clarify what you mean by "6 aspects"? line 90 - what does "top rated" provincial hospitals mean? line 98 - trained interviewers helped illiterate outpatients - how many were included in the study? line 99 - what do you mean by qualified investigators? line 111 - related to my comment above - it is not clear how patients were recruited - how were individuals approached to participate? was it everyone in the waiting room? randomly chosen? line 116 - what is unified guidance language? Table 2 - why is the correct answer to "Do you think a GP is the same as a PCP" = no? Aren't they the same? This goes back to one of my points above to clearly explain what Chinese term was used in the survey for "GP" as various terms are used in China. line 162 - rephrase sentence - grammar Table 3's heading is not clear. isn't the main outcome here people who have never heard of GPs?
--

VERSION 1 – AUTHOR RESPONSE

Editorial Requests:

- Please revise your title. We ask authors to refrain from using declarative titles (i.e those that state the study's main findings). Please amend your title so that it frames the research question and includes the research design and setting.

Response: We have made the change accordingly.

- Please elaborate on the development of the survey instrument. Was it developed specifically for this study, or has it been used in previous research? Was it pilot tested?

Response: Based on the Chinese medical students' textbook called Introduction to General Practice (Zhu S.Z. Introduction to General Practice (Fourth Edition). Beijing: People's Medical Publishing House, 2013. or in Chinese: 祝墀珠. 全科医学概论.第4版[M]. 人民卫生出版社, 2013.).

Revised text in the Methods section: "The questionnaire was designed based on the Introduction to General Practice 14, and with the help of scholars familiar with the topic with final review by clinical experts in the field (i.e., university hospital instructors of GPs). Surveys were pre-tested prior to study implementation to ensure that potential participants understood the wording and meaning of study questionnaires."

- Please improve the data sharing statement on page 12. How can other researchers access the dataset underlying the results reported in this manuscript? For example, is it available upon request from the corresponding author? Are there any ethical or legal restrictions to accessing the dataset?

Response: The data are available upon request from the corresponding author. Data requesters need to provide their research objective, design, and ethical approval documents. See the revised data sharing statement section.

- The quality of English could be improved in places e.g. page 2: "The targeted individuals were asked to answer the questionnaires independently and anonymously to avoid the potential bias." Please copy-edit the paper, consulting a native English speaker (if possible).

Response: We have reviewed the manuscript and corrected the highlighted issue and other instances.

Reviewers' Comments to Author:

Reviewer: 1

Reviewer Name: Martin Roland

Institution and Country: University of Cambridge, UK

Competing Interests: None

Some years ago, China developed a national policy to provide general practice based primary care and this has been implemented to varying extents across the country. However, patients still preferentially go to hospital clinics and primary care facilities may be under-utilised.

The paper explores some of the reasons why – including that many patients had never heard of a GP and had no idea what they did. To this extent the paper is useful. It's clearly set out and could be published. The survey achieved a very high response rate of 94% (these types of response rate are hardly ever seen outside China).

RESPONSE:

We appreciate this comment and agree that the response rate was high, as is not unusual in China—as noted by the reviewer.

One issue which is not possible to deduce from the paper is whether the patients were clear what they were being asked about. So the reader doesn't know what term was used for GP, Primary Care Physician or Family Practice and whether respondents could have been confused by the terms used. The best way to address this would be to have the questionnaire survey reviewed by someone bilingual in English and Mandarin or whichever local dialect was used. Other than that, the analyses appear sound and the results clearly presented.

RESPONSE:

We agree that more clarity was necessary about the survey and possible interpretations by the respondents. We addressed this comment and that of Reviewer # 2 in the Methods section, specifically adding greater clarity to the terminology and methods to ensure respondents were able to understand what was asked, while realizing that with any survey some respondents may interpret questions unexpectedly. Please see the revised Methods section.

Text added to the Methods section:

“The questionnaire was designed based on the Introduction to General Practice 14, and with the help of scholars familiar with the topic with final review by clinical experts in the field (i.e., university hospital instructors of GPs). Surveys were pre-tested prior to study implementation to ensure that potential participants understood the wording and meaning of study questionnaires.”

What for me is missing from the paper is to locate it in the national policy context and, in particular, the local implementation of that policy in Jianxi province – for example, how many GP are there, how evenly are they spread, how long have they been there etc.

RESPONSE:

We have added more context (i.e., local implementation of the GP policy in Jianxi province) to the text in the Discussion section.

Other than that, I think this is an interesting and useful paper and could potentially be published.

RESPONSE:

We agree with the reviewer that the topic is of interest and can be of use to policy makers and other key stakeholders.

Reviewer: 2

Reviewer Name: Jason Ong

Institution and Country: London School of Hygiene and Tropical Medicine, Monash University, Australia

Competing Interests: None declared

Thank you for the opportunity to review your interesting and well-written research. Some clarification points to improve your manuscript.

MAJOR ESSENTIAL CHANGES

line 27 - this is your major result that half of respondents have not heard of GP. This may also be influenced by the the Chinese phrase used for GP - did you use a technical term that most lay people would not understand? please clarify how this question was phrased in your methods section.

RESPONSE:

A similar comment was introduced by Reviewer # 1 and we have expanded a discussion of this in the Methods section, as the reviewer suggests.

Please see the revised text in the Methods section: “In China, general practitioners are typically referred to as ‘GPs’ or family practitioners (FPs). However, GPs (or FPs) in China are not the same as primary care physicians (PCPs). In China, PCPs are also referred to as ‘barefoot doctors’ (in Chinese ‘Chijiao doctors’ or ‘Tongke practitioners’). PCPs (or barefoot doctors) are given basic training in western disease control and traditional Chinese medicine or ethnic medicine and typically have limited technical skills or medical capability 14. China developed the national GP policy to provide general practice based primary care, and this has been implemented to varying extents across the country. The Chinese government continues to train qualified GPs with the goal of replacing existing PCPs (or barefoot doctors).”

line 92 - what spatial sampling was used?

RESPONSE:

Spatial sampling means participants in different floors and different directions of the outpatient building were selected.

We added the following to the Methods section:

“Convenience sampling and spatial sampling (i.e., targeting different floors and directions to improve recruitment of potential participants) were used to collect the data.”

line 94 - it is not clear what type of outpatients has been recruited. this sentence says "general outpatient hall" - can you please expand?

RESPONSE:

All outpatients were initially targeted, with those agreeing to participate included. The tertiary hospitals are very large, and there are different buildings (e.g., the outpatient building, different surgical buildings, emergency building). There was one outpatient building in each hospital, and was commonly known as the "general outpatient hall". We removed the previous text to avoid confusion and to be more specific (i.e., surveys were collected throughout the outpatient building).

Revised text:

"Questionnaires were given to the registered outpatients who were waiting to see a doctor in different floors and different directions of the outpatient building."

line 94 - those who "qualified" - what were your inclusion/exclusion criteria?

RESPONSE:

Outpatients with a registered physician in provincial tertiary hospitals were invited to take part in a survey, while those referred from other health service institutions were excluded. See the 2.1. Participants section.

We have added more detail to the Methods section regarding inclusion/exclusion criteria.

Revised text: "Those who qualified (i.e., including those with a registered physician in a provincial tertiary hospital, while excluding those referred from another health service institutions) were informed of the study and asked if they were willing to participate establishing informed consent."

line 105 - how was chronic disease status defined?

RESPONSE:

We defined chronic disease as self-reported 'yes' to the question of having chronic disease. We included the text "self-reported" when defining this in the Methods section.

line 119 - it is not clear how your multivariate model was built? did you use backward elimination? another technique?

RESPONSE:

Our multivariable model included demographic variables as indicated in the Methods section. Model selection was not automated (i.e., we did not employ backward or stepwise or other techniques) so that all the statistical values of independent variables were reported. We did this because we want to present the full information to our potential readers.

what stats package was used?

RESPONSE:

We used SPSS as indicated in the Methods section "Statistical Package for the Social Sciences (SPSS) version 22.0 for Windows (SPSS Inc., Chicago, IL, USA) was employed to run all statistical analysis."

Discussion - more detail is needed regarding how your results compare with other published literature in the Chinese context

RESPONSE:

We have expanded the Discussion and included more relevant citations and how this compares to our study. See the revised Discussion section.

Revised text:

"According to the 2016 Statistical Yearbook of China's National Health and Family Planning Commission 19, the total number of licensed GPs and assistant GPs was just 5.3% of the total number of physicians in China. The General Office of the State Council promulgated the Opinion on Promoting the Construction of Hierarchical Medical Treatment System 20, which aims to achieve a goal of having 2-3 qualified GPs for every 10,000 citizens. However, the average number of GPs per 10,000 citizens in China was just 1.27 overall, with 1.71 per 10,000 in more developed provinces in the eastern coastal areas, with 0.91 and 0.99 per 10,000 in less developed areas in central and western regions, respectively. Jiangxi Province is a typical central province in China with 0.54 GPs per 10,000 permanent residents in 2016, among which, most of the licensed GPs were specialists,

leaving few who were truly trained by the standard procedures of GPs providing non-specialist care 21. Moreover, nearly half of the GPs did not work in primary care facilities, giving way to a large gap in primary care facilities which desperately need GPs 21-22.”

MINOR ESSENTIAL CHANGES

Line 21 - rephrase.. .the study is not about 'GPs in China's tertiary hospitals' - the last part of the sentence refers to outpatient attendees line 23 - add whether the 565 outpatients were sequentially/randomly chosen, from any particular speciality outpatients?

RESPONSE:

We have made this revision.

line 71 - please clarify what you mean by "6 aspects"?

RESPONSE:

We have defined this in the text.

Revised text: “Integration of Six Aspects” in primary care facilities refers to the community health service network system which integrates community prevention, health care, medical treatment, rehabilitation, health education and family planning technology guidance. The comprehensive function of the concept meets various requirements in health care 12-13.”

line 90 - what does "top rated" provincial hospitals mean?

RESPONSE:

We have defined this in the text as “top rated (e.g., largest hospitals with high levels of resources and high-quality care)”.

line 98 - trained interviewers helped illiterate outpatients - how many were included in the study?

RESPONSE:

We did not track this measure. We have responded to this comment by adding greater detail in the Limitations section.

Revised text: “Given the pragmatic nature of data collection in hospital settings, we did not track how many surveys were done through face-to-face interviews versus those where individuals completed the surveys by themselves or those where certain questions were missing and participants were asked to complete missing items.”

line 99 - what do you mean by qualified investigators?

RESPONSE:

We have responded to this comment by adding greater detail in the Methods section.

Revised text: “Lastly, completed questionnaires were checked by qualified investigators (i.e., graduate students specifically trained to carryout data collection for this study) to ensure the completeness of the questionnaires with immediate follow-up with participants needing further information, as needed.”

line 111 - related to my comment above - it is not clear how patients were recruited - how were individuals approached to participate? was it everyone in the waiting room? randomly chosen?

RESPONSE:

We have responded to this comment by adding greater detail in the Methods section.

Revised text: “Convenience sampling on different floors and throughout the outpatient building was carried out.”

line 116 - what is unified guidance language?

RESPONSE:

We have responded to this comment by adding greater detail in the Methods section. Essentially we only meant that the study was clearly explained in a consistent or unified manner. That said, we have reframed the statement to increase clarity.

Revised text: “For those who could not complete survey questions independently, we employed a rigorously trained and qualified staff of investigators to interview them face-to-face by using a consistent and clear explanation of the purpose and significance of the study before the survey was conducted while at the same time answering questions raised by potential participants as they were encountered.”

Table 2 - why is the correct answer to "Do you think a GP is the same as a PCP" = no? Aren't they the same? This goes back to one of my points above to clearly explain what Chinese term was used in the survey for "GP" as various terms are used in China.

RESPONSE:

We have responded to this comment by adding greater detail in the Methods section.

Revised text: "In China, general practitioners are typically referred to as 'GPs' or family practitioners (FPs). However, GPs (or FPs) in China are not the same as primary care physicians (PCPs). In China, PCPs are also referred to as 'barefoot doctors' (in Chinese 'Chijiao doctors' or 'Tongke practitioners'). PCPs (or barefoot doctors) are given basic training in western disease control and traditional Chinese medicine or ethnic medicine and typically have limited technical skills or medical capability 14. China developed the national GP policy to provide general practice based primary care, and this has been implemented to varying extents across the country. The Chinese government continues to train qualified GPs with the goal of replacing existing PCPs (or barefoot doctors)."

line 162 - rephrase sentence – grammar

RESPONSE:

We have made this edit.

Table 3's heading is not clear. isn't the main outcome here people who have never heard of GPs?

RESPONSE:

We have made this edit.

VERSION 2 – REVIEW

REVIEWER	Martin Roland University of Cambridge UK
REVIEW RETURNED	02-Jan-2018

GENERAL COMMENTS	I think that the authors have satisfactorily addressed the comments that I made on the previous version of the paper.
---

REVIEWER	Jason Ong London School of Hygiene and Tropical Medicine
REVIEW RETURNED	17-Jan-2018

GENERAL COMMENTS	Thank you for answering my comments comprehensively. I have no further comments and wish you the very best as you continue to research about the role of primary care in China.
---

VERSION 2 – AUTHOR RESPONSE

Editorial Comments:

Major

- We have some concerns about the ethics approval procedure for this study. We would normally require approval from the relevant local ethics committee(s) in the country where the data collection took place. We note that the study was approved by the IRB of the Texas A&M University Health Science Center but it is not clear whether you also obtained approval from the ethics committees of the hospitals where the data was collected from. Please clarify this.

Response: To address the editor's note of concern, study procedures were already in place and approved with both Universities prior to the study being conducted in accordance with all proper procedures. We appreciate your pointing out the oversight of not including the complete approval information in the text. We have ensured the text contains all appropriate Institutional Review Boards.

Please see the revised text : "This study was conducted in accordance with the Declaration of Helsinki. Local institutional protocols at Wuhan University were also followed to protect participants' confidentiality, with ethics approval granted by Wuhan University's School of Public Health (IRB Number: J-17-2016). Informed consent was obtained from all participants. Hospitals are considered public places in China, and approval from the ethics committees of the hospitals was not required. The Institutional Review Board of Texas A&M University (IRB Number: IRB2016-0285D) also reviewed and approved of study protocols."

Minor

- The title you have provided is still declarative i.e. you are stating that outpatient populations have a limited awareness of the role of GPs. Can you please amend to, for example: "Awareness of the role of general practitioners in primary care among outpatient populations: evidence from a cross-sectional survey of tertiary hospitals in China."

Response: This change has been made, as suggested.

Revised title: "Awareness of the role of general practitioners in primary care among outpatient populations: evidence from a cross-sectional survey of tertiary hospitals in China."

- The data sharing statement on page 12 still states: "Data are available to the corresponding author and research team only." In your rebuttal letter you said that data was available upon request from the corresponding author and that this statement had been amended. Can you please amend this as previously indicated?

Response: We have amended this as previously indicated.

Revised text in this section: "Data are available upon request from the corresponding author. Data requesters are required to provide their research objective, design, and ethical approval documents."

- The quality of English still needs improving in places. There are still typos present e.g. abstract: "This is cross-sectional survey study." should be "This is a cross-sectional survey study." See also: "This study adds scientific rationale that.." (abstract) is unclear. The manuscript should be carefully proofread.

Response: We have made the changes in the text and reviewed the manuscript.

Reviewers' Comments to Author:

Reviewer: 1

Reviewer Name: Martin Roland

Institution and Country: University of Cambridge, UK

Competing Interests: None declared

I think that the authors have satisfactorily addressed the comments that I made on the previous version of the paper.

RESPONSE:

We thank the reviewer for the valuable review comments throughout the entire process.

Reviewer: 2

Reviewer Name: Jason Ong

Institution and Country: London School of Hygiene and Tropical Medicine, UK

Competing Interests: None declared